# Validation of an Equine Smart Textile System for Heart Rate Variability: A Preliminary Study

**DOI:** 10.3390/ani13030512

**Published:** 2023-02-01

**Authors:** Persephone McCrae, Hannah Spong, Nadia Golestani, Amin Mahnam, Yana Bashura, Wendy Pearson

**Affiliations:** 1Department of Animal Biosciences, University of Guelph, Guelph, ON N1G 2W1, Canada; 2Department of Research and Development, Myant Inc., Toronto, ON M9W 1B6, Canada

**Keywords:** electrocardiogram, cardiology, equine, smart textile, textile computing, heart rate variability, arrhythmia

## Abstract

**Simple Summary:**

Electrocardiography and heart rate variability allow for greater understanding of equine cardiovascular health and fitness. Devices required to obtain these data are typically cumbersome, limiting their use to clinicians. The objective of this study was to validate a user-friendly smart textile system against the gold-standard telemetric device. Electrocardiograms (ECGs) were obtained simultaneously using both devices in horses at rest and undergoing submaximal exercise. We did not observe any differences in metrics of heart rate or heart rate variability, indicating that the smart textile system is a reliable alternative device for ECG monitoring of horses during rest and submaximal exercise.

**Abstract:**

Electrocardiograms (ECGs), and associated heart rate (HR) and heart rate variability (HRV) measurements, are essential in assessing equine cardiovascular health and fitness. Smart textiles have gained popularity, but limited validation work has been conducted. Therefore, the objective of this study was to compare HR and HRV data obtained using a smart textile system (Myant) to the gold-standard telemetric device (Televet). Simultaneous ECGs were obtained using both systems in seven horses at rest and during a submaximal exercise test. Bland–Altman tests were used to assess agreement between the two devices. Strong to perfect correlations without significant differences between the two devices were observed for all metrics assessed. During exercise, mean biases of 0.31 bpm (95% limits of agreement: −1.99 to 2.61) for HR, 1.43 ms (−11.48 to 14.33) for standard deviation of R-R intervals (SDRR), and 0.04 (−2.30 to 2.38) for the HRV triangular index (TI) were observed. A very strong correlation was found between the two devices for HR (*r* = 0.9993, *p* < 0.0001) and for HRV parameters (SDRR *r* = 0.8765, *p* < 0.0001; TI *r* = 0.8712, *p* < 0.0001). This study demonstrates that a smart textile system is reliable for assessment of HR and HRV of horses at rest and during submaximal exercise.

## 1. Introduction

Arrhythmias are a common occurrence in equine medicine, encompassing a wide range of severity [1]. Arrhythmias may be physiological and pathological [1]. Physiological arrhythmias tend to disappear during periods of stress, such as exercise [1,2]. Pathological arrhythmias, however, are typically the result of an underlying disease and are associated with poor performance, an increased likelihood of acute cardiac injury, and even death [1,3]. Sudden cardiac death (SCD) is a subset of sudden death in horses, in which a presumedly healthy horse experiences acute collapse or death during or immediately following exercise [4]. A 2010 study of 268 Thoroughbred racehorses found that 56% of definitive diagnostic cases and 54% of presumptive diagnostic cases were due to cardiac and/or pulmonary failure [5]. Due to the prevalence of cardiac arrhythmias and SCD in horses, it is essential that performance horses are proactively monitored. Clinical signs and symptoms are not always present in horses with arrhythmias; therefore, electrocardiograms (ECGs) are required to make a definitive diagnosis [6].

Heart rate variability (HRV) describes the changes between consecutive cardiovascular cycles [7]. Cardiovascular monitoring has become more accessible in human and veterinary medicine, particularly due to the growth in wearable technologies. While HRV has previously been used to investigate changes in vagosympathetic balance, there is increased interest in utilizing HRV to identify arrhythmias, such as atrial fibrillation [7,8,9]. Heart rate variability data can be obtained from telemetric ECGs, heart rate (HR) monitors, and other technologies that record pulse waveform data [7]. While HR monitors are easy to use, the QRS-T morphology cannot be visualized for further assessment, and many of these devices apply heavy filtering techniques that may exclude arrhythmic beats [7]. Additionally, telemetric ECG devices can be challenging to use and are prone to motion artifacts (MAs), which can limit analysis and lead to misinterpretation [10,11]. An equine fitness tracker for daily exercise monitoring (Equimetre) was recently validated for HR, HRV, and arrhythmia detection, indicating that excellent-quality data can be obtained from user-friendly devices [12,13]. Recent advancements in smart textile technologies have resulted in increased interest in their applications within human [14,15] and veterinary medicine [16]. Previous work has compared smart textile electrodes to standard adhesive silver/silver chloride (Ag/AgCl) electrodes in horses at rest [17,18] and during treadmill exercise [19]. Textile electrodes have been found to perform as well as, or better than, Ag/AgCl electrodes, indicating that smart textiles are a reliable alternative to standard cardiovascular monitoring techniques [17,18,19]. However, variable signal quality has been observed, potentially due to differences in conductive yarn properties. We have previously shown that excellent-quality ECGs, with much lower rates of MAs than had previously been reported in horses [17,19], can be obtained using a smart textile girth band with electrodes composed of silver- and carbon-coated yarns [18]. Since this study utilized the same recording device without a gold-standard device, comparisons were only made between electrode types, and not between devices. Additionally, the data were collected only from horses at rest, without any structured exercise and no metrics of HRV were assessed. Therefore, the objective of the present study was to validate a smart textile system against the gold-standard telemetric device for measures of HRV in horses at rest and during submaximal exercise.

## 2. Materials and Methods

### 2.1. Animals and Housing Conditions

Seven mixed-breed, English performance horses enrolled in full-time showjumping training were recruited for this study (median age = 13 years (IQR: 11.5–14 years); 4 mares and 3 geldings; 4 Warmblood/Warmblood crosses, 2 Thoroughbreds, and 1 Thoroughbred/Clydesdale cross). The study was performed in accordance with the University of Guelph Animal Utilization Protocol (reference No. 4705). Written informed consent was obtained from horse owners (or an authorized agent for the owner). Horses were housed in 12 × 12 ft box stalls with ad libitum access to hay and water.

### 2.2. Data Collection

Horses were instrumented with a smart textile girth band (Skiin Equine, Myant Inc., Toronto, ON, Canada; Figure 1) and a telemetric ECG device (Televet 100, Jørgen Kruuse, Denmark). The smart textile band had five 4 cm square electrodes composed of silver- and carbon-coated yarns, arranged in a modified base-apex configuration, as previously described [18]. The Ag/AgCl electrodes were placed underneath the textile band, against the skin, and directly above the textile electrodes, to maintain continuity between the three leads, without electrode overlap. The skin was not prepared, and the hair was not shaved prior to application of either electrode type. Both the textile and Ag/AgCl electrodes had a salt- and chloride-free, electrically conductive gel (Spectra 360 Electrode Gel, Parker Laboratories Inc., Fairfield, NJ, USA) applied. A surcingle was mounted on top of the textile band, with the Televet recording device was attached to it.

For both systems, standard ECG cables connected the electrodes to the recording device. The smart textile system recorded ECG data at a sampling frequency of 320 Hz, while the Televet system recorded at a sampling frequency of 500 Hz. Data from both systems were transmitted via Bluetooth to either a mobile phone or laptop during recording.

Electrocardiograms were recorded simultaneously using the two devices during both rest and exercise. Rest measurements were taken prior to exercise for a total of 15 min. Horses were unrestrained within stalls and were permitted to freely move about and eat during the recording. Following the completion of the data collection at rest, horses were led to a large, sand arena for the exercise test. Horses were fitted with a lunging cavesson and exercised on a 20 m circle by the same handler. Activity data were collected using a field exercise test previously developed for dressage and showjumping horses [20]. This protocol includes 2 min trot (head loose), 2 min trot (outline and working), 2 min canter, and 2 min extended canter [20]. Since this test was developed for horses undergoing ridden exercise and the present study exercised horses without a rider, an additional minute was added to each step of the test to ensure that there was sufficient time for the handler to achieve the desired gait, speed, and outline of the horse. Horses were walked in hand prior to and at completion of the standardized exercise test to obtain data for walk and post-exercise walk (active recovery) for 5 min each.

### 2.3. Data Analysis

In total, 14 ECG recordings were obtained, with one recording per device per horse. Raw data were exported from both devices and imported into an HRV software (Kubios HRV Premium, version 3.5.0) for analysis. Data were not filtered during analysis. Automatically identified R-peaks and data from the same channel were manually assessed by one operator to correct any misidentified or missed peaks. The data were segmented to obtain HR and HRV metrics for rest, walk, trot, canter, and post-exercise walk individually.

Time-domain metrics were determined for each device and activity, including mean heart rate (HR), the standard deviation of R-R intervals (SDRR), the HRV triangular index (the integral of the density of the R-R interval histogram divided by its height—HRV TI), the triangular interpolation of R-R interval histogram (TIRR), and the root mean square of successive R-R interval differences (RMSSDs). Additionally, nonlinear measurements were obtained from the Poincaré plot by fitting an ellipse to the plotted points. The standard deviation perpendicular to the line of identity (SD1) and the standard deviation along the line of identity (SD2) of the Poincaré plot were determined, where SD1 correlates with short-term HRV and SD2 correlates with long-term HRV [9,12,21].

### 2.4. Statistical Analysis

All data were assessed for individual activities (rest, pre-exercise walk, trot, canter, post-exercise walk) and pooled for rest and exercise. Data were assessed for normality using Shapiro–Wilk tests. A Pearson correlation coefficient was calculated for both devices, where an *r* value of 1.0 was considered to be a perfect correlation, an *r* value of 0.8 to 1.0 was considered to be a very strong correlation, and an *r* value of 0.6 to 0.8 was considered to be a strong correlation. Bland–Altman analysis was used to assess agreement of each metric between the two devices. Paired *t*-tests were used to determine if the bias between the two devices was significantly different from 0. GraphPad Prism (9.4.1) was used to perform all statistical analyses with significance set at *p* ≤ 0.05. 

## 3. Results

In total, seven pairs of synchronous ECG recordings (one from each device) were obtained (Figure 2). One horse’s data were excluded from the exercise portion due to overexcitement and bucking on the lunge line, which did not appear to be due to either device (*n* = 6). All other horses tolerated the smart textile and Televet systems well during both rest and exercise. The horses appeared comfortable with both systems and no behavioural changes or skin irritation were observed.

Heart rate and HRV metrics are listed for each activity in Table 1. When data were divided by activity, strong to perfect correlations were found between the two devices for all metrics (Table 2). Bland–Altman analysis revealed excellent agreement between the two devices for all metrics assessed. Paired *t*-tests revealed no significant differences between the two devices for any of the metrics assessed for individual activities.

### 3.1. Heart Rate

Mean HR was found to be perfectly correlated at rest (*r* = 1.0, *p* < 0.0001) and very strongly correlated during exercise (all activities pooled: *r* = 0.9993, *p* < 0.0001) (Figure 3A,B). Biases of −0.0007 bpm (95% limits of agreement: −0.0217 to 0.01965 bpm, *p* = 0.86) and 0.3107 bpm (−1.993 to 2.614 bpm, *p* = 0.21) were found at rest and during exercise, respectively (Figure 3C,D).

### 3.2. Heart Rate Variability

The SDRR was found to be very strongly correlated at rest (*r* = 0.9997, *p* < 0.0001) and during exercise (*r* = 0.8765, *p* < 0.0001), with biases of 0.0094 ms (−0.8832 to 0.9021 ms, *p* = 0.96) and 1.425 ms (−11.48 to 14.33 ms, *p* = 0.30) for rest and exercise, respectively (Figure 4A,B).

The HRV TI and TIRR were both very strongly correlated at rest (TI: *r* = 0.8087, *p* < 0.0138; TIRR: *r* = 0.9999, *p* < 0.0001) and during exercise (TI: *r* = 0.8693, *p* < 0.0001; TIRR: *r* = 0.8712, *p* < 0.0001). Small biases were found for HRV TI at rest (−0.4024 (−5.575 to 4.771), *p* = 0.71) and during exercise (0.0406 (−2.303 to 2.384), *p* = 0.87) (Figure 4C,D). Small biases were also observed for TIRR (rest: −0.1429 ms (−30.66 to 30.38 ms), *p* = 0.98; exercise: −2.659 ms (−69.28 to 63.97 ms), *p* = 0.71) (Figure 4E,F).

The RMSSD was very strongly correlated at rest (*r* = 0.9999, *p* < 0.0001) and during exercise (*r* = 0.9601, *p* < 0.0001), with small biases of 0.0713 ms (−0.5503 to 0.6928 ms, *p* = 0.57) and 0.9692 ms (−4.617 to 6.555 ms, *p* = 0.11) for rest and activity, respectively (Figure 5).

Correlations were very strong for SD1 and SD2 at rest (SD1: *r* = 0.9999, *p* < 0.0001; SD2: *r* = 0.9998, *p* < 0.0001) and during exercise (SD1: *r* = 0.9572, *p* < 0.0001; SD2: *r* = 0.9032, *p* < 0.0001). Analysis yielded biases of 0.0497 ms (−0.3927 to 0.4921 ms, *p* = 0.58) and 0.1184 ms (−0.6844 to 0.9212 ms, *p* = 0.47) for SD1 and SD2 at rest, respectively. Biases of 0.7733 ms (−3.395 to 4.942 ms, *p* = 0.10) and 0.2946 ms (−9.681 to 10.27 ms, *p* = 0.78) were found for SD1 and SD2 during activity, respectively (Figure 6).

## 4. Discussion

Overall, the results of this preliminary study indicate excellent agreement between the smart textile system and the gold-standard telemetric device in horses at rest and during submaximal exercise. Strong to perfect correlations between the two devices were observed for HR and metrics of HRV. There were no significant differences observed between both systems at all tested gaits (walk, trot, canter), indicating the potential of smart textiles in ECG and HRV monitoring of exercising horses.

Heart rate variability has been broadly studied in horses since the 1990s [8,9,12,22,23,24,25,26,27,28,29,30,31,32,33,34,35]. A 2018 study of 14 horses that presented with atrial fibrillation found that HRV data, in particular short-term variability (RMSSD), could be used to differentiate between horses with atrial fibrillation and sinus rhythm [8]. Furthermore, shortening of the R-R intervals during exercise greater than 6% is representative of arrhythmias, and may allow for increased automation of arrhythmia detection in horses [9]. Analysis of HRV data can also provide insight into fitness and the efficacy of training programs, where changes in HRV have been found within different competition periods [36], in response to overtraining [30], and in determining if horses are unfit prior to competition [33]. In addition, researchers have found that HRV analysis can detect minute differences in workload compared to HR alone [37]. Furthermore, changes in HRV have been shown to be useful in assessing pain and welfare. Becker-Birck et al. found that horses displayed increased HR and decreased HRV prior to a competition, which is a sign of anticipatory stress [28]. In laminitic horses, HRV was suggested as an alternative to blood sampling for the assessment of pain [24].

Recent technological advancements have led to an increase in devices for equine veterinarians, researchers, and owners/trainers. Indeed, the Televet system is considered to be the gold standard in equine cardiovascular monitoring during rest and activity. However, its reliance on individually placed adhesive electrodes limits its usage to clinicians, primarily. As interest increases in the remote monitoring of health, it is important that user-friendly devices that can be incorporated into routine care and exercise are developed and validated. Smart textiles have been developed as an alternative to standard Holter devices, as they are accessible and comfortable [38]. Horses have been shown to exhibit an increase in HRV due to the stress of restraint during procedures, such as application of standard ECG devices [39]. Therefore, utilizing a smart textile product may allow for greater comfort and reduced stress of animals during monitoring. Textile electrodes in wearable systems allow for remote, long-term monitoring of patients, resulting in reduced cost and improved efficiency of diagnosis and subsequent treatment [40]. Furthermore, textile electrodes are typically integrated into garments, which allows for ease of use of use, without requiring specialized training to place electrodes at anatomical landmarks. Previously, the functionality of smart textiles has been investigated in the horse by one other research group. In horses, when unrestrained at rest, the authors observed a significant reduction in the percentage of the signal impacted by MAs when ECGs were obtained using textile electrodes, as compared to adhesive Ag/AgCl electrodes, with MA rates of 35% and 51% for textile and Ag/AgCl, respectively [17]. More recently, we have observed rates of MAs to be below 0.5% in unrestrained horses at rest using a different smart textile system, indicating that textile electrodes may be less prone to MAs [18]. We did not quantify MAs in the present study, as the aim was to compare measures of HRV. Therefore, further work on the presence of MAs obtained using textile systems, particularly during exercise, is warranted. Felici et al. compared ECG signal quality between data obtained using textile electrodes and Ag/AgCl electrodes during treadmill exercise at a walk, trot, and gallop [19]. The authors observed no differences in signal quality between the two electrode types when the adhesive electrodes were applied without glue; however, the quality was variable across gaits and time, with significant worsening as the test progressed [19]. To date, no HRV parameters have been reported using smart textile systems in the horse. However, good agreement of HRV between a smart textile system and the Televet has been shown in standing, non-sedated sheep [41]. The two devices were observed to be highly correlated for mean R-R intervals (*R^2^* = 0.96), percentage of successive R-R intervals that differ by more than 50 ms (pNN50; *R^2^* = 0.95), RMSSD (*R^2^* = 0.75), and SD1 (*R^2^* = 0.75), with variable degrees of correlation for the index of sympathovagal balance (LF/HF; *R^2^* = 0.30), sample entropy (SampEn; *R^2^* = 0.47), SD2 (*R^2^* = 0.55), and SDRR (*R^2^* = 0.63) [41]. In humans, good agreement between smart textile shirts and standard ECG devices has been observed during cycling [42,43], walking, and running on a treadmill [44]. These studies, along with the present study, indicate that smart textiles can be utilized to obtain reliable HR and HRV data in a variety of species.

## 5. Conclusions

This preliminary study indicates that excellent-quality HR and HRV data can be obtained using a smart textile band in horses during rest and submaximal exercise. Further work with a greater number of horses and maximal-intensity exercise is warranted.

## Figures and Tables

**Figure 1 animals-13-00512-f001:**
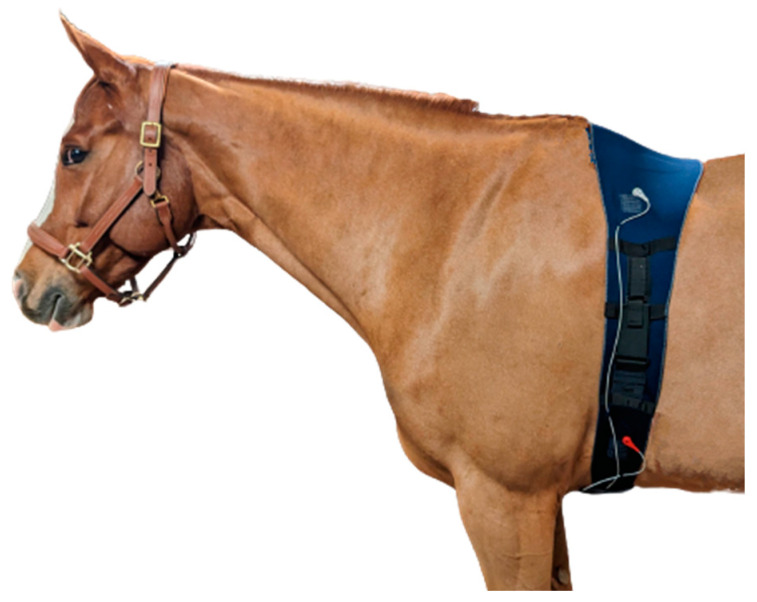
The smart textile girth band system (Myant Inc., Skiin Equine). Light-blue squares indicate electrode positions.

**Figure 2 animals-13-00512-f002:**
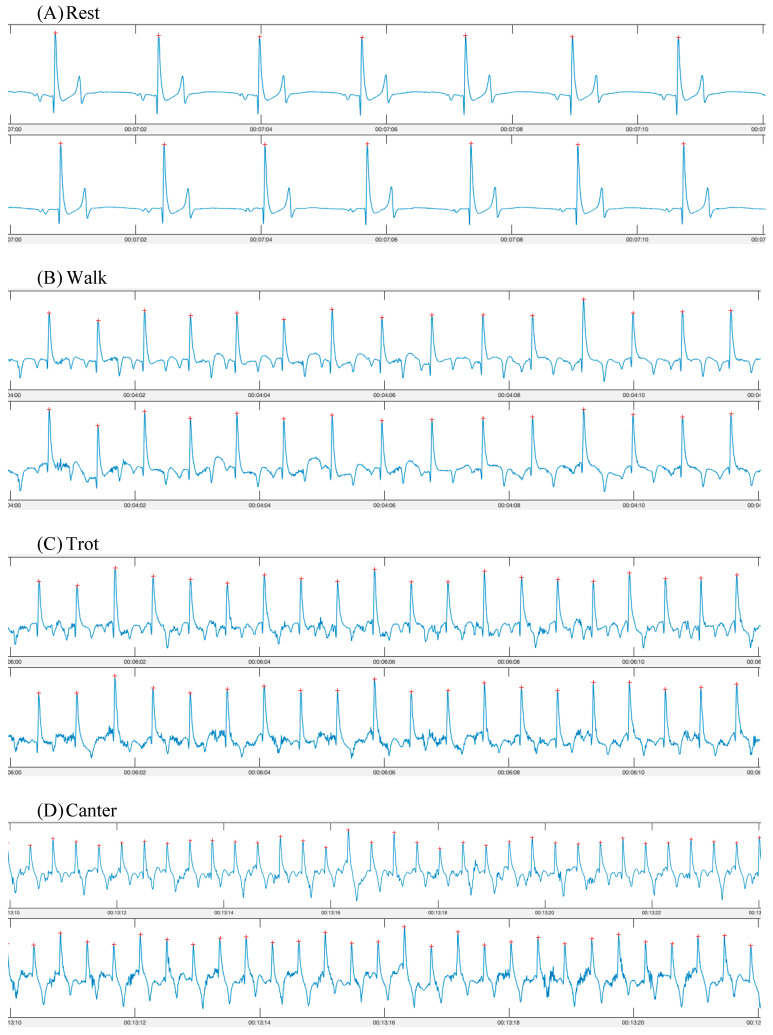
ECG traces obtained using a smart textile device (Myant—top row) and a standard telemetric device (Televet—bottom row). ECGs were obtained simultaneously during (**A)** rest; (**B**) walk; (**C**) trot; (**D**) canter.

**Figure 3 animals-13-00512-f003:**
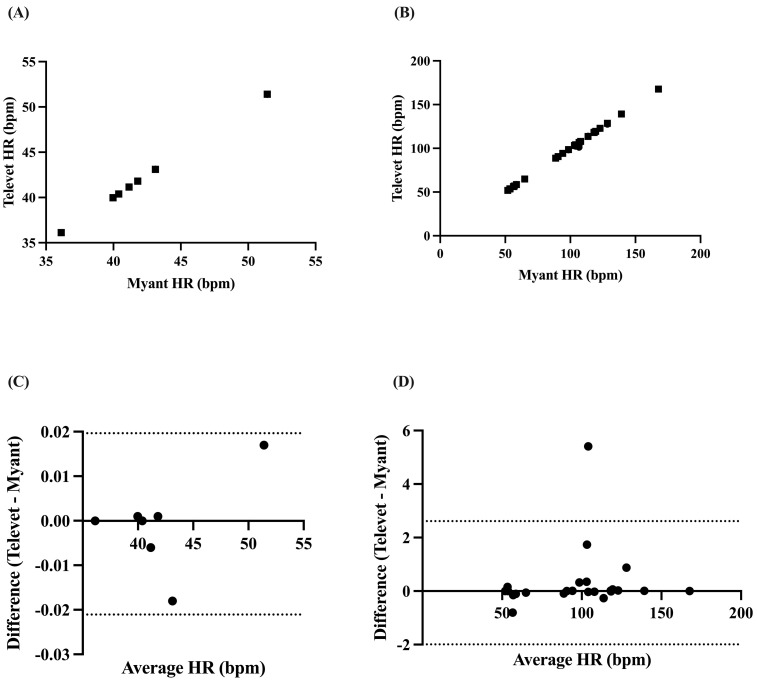
Correlation between two ECG devices (Myant, Televet) for mean HR at rest (**A**) and during exercise (**B**). Bland–Altman plots demonstrating the agreement between the two devices for mean HR at rest (**C**) and during exercise (**D**), where x-axes are the average of HR for each device, y-axes are the difference of Televet minus Myant HR, solid lines indicate bias, and dashed lines indicate 95% limits of agreement.

**Figure 4 animals-13-00512-f004:**
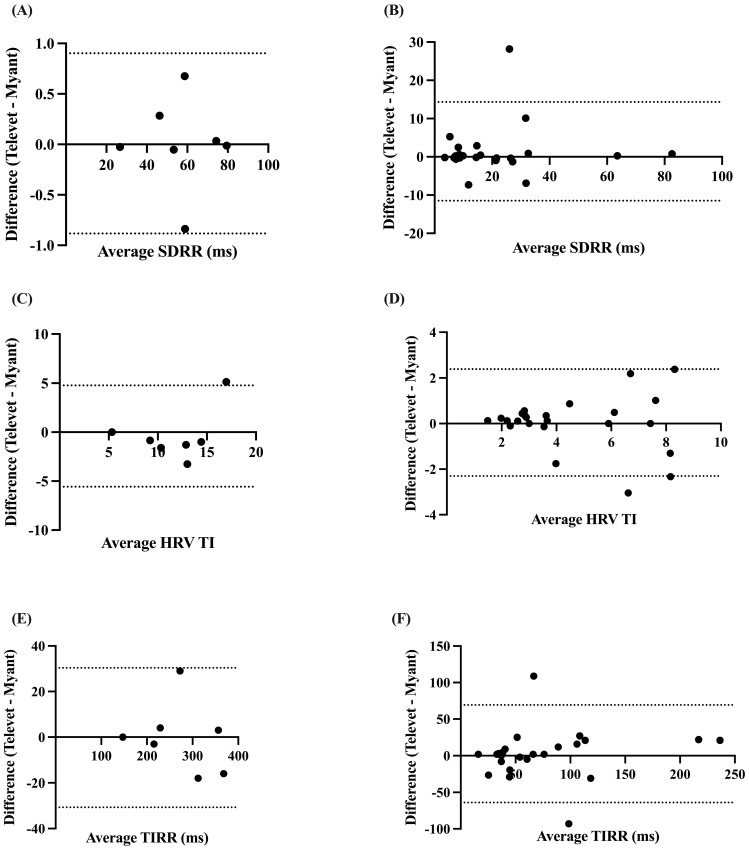
Bland–Altman plots demonstrating the agreement between the two devices for HR metrics at rest (**A**,**C**,**E**) and during exercise (**B**,**D**,**F**). SDRR = standard deviation of R-R intervals; HRV TI = heart rate variability triangular index; TIRR = triangular interpolation of the R-R interval histogram. X-axes are the average of each metric for each device; y-axes are the difference of Televet minus Myant for each metric. Solid lines indicate bias and dashed lines indicate 95% limits of agreement.

**Figure 5 animals-13-00512-f005:**
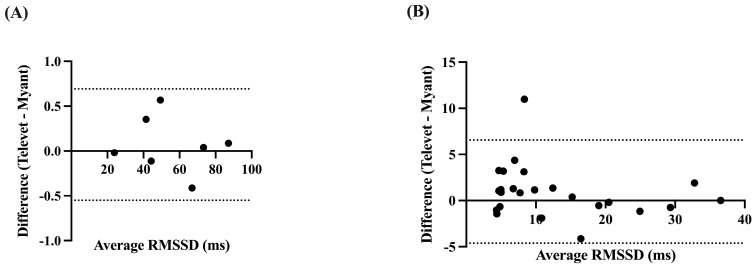
Bland-Altman plots demonstrating the agreement between the two devices for root mean square of successive R-R interval differences (RMSSD), at rest (**A**) and during exercise (**B**). X-axes are the average of RMSSD for each device; y-axes are the difference of Televet minus Myant RMSSD. Solid lines indicate bias and dashed lines indicate 95% limits of agreement.

**Figure 6 animals-13-00512-f006:**
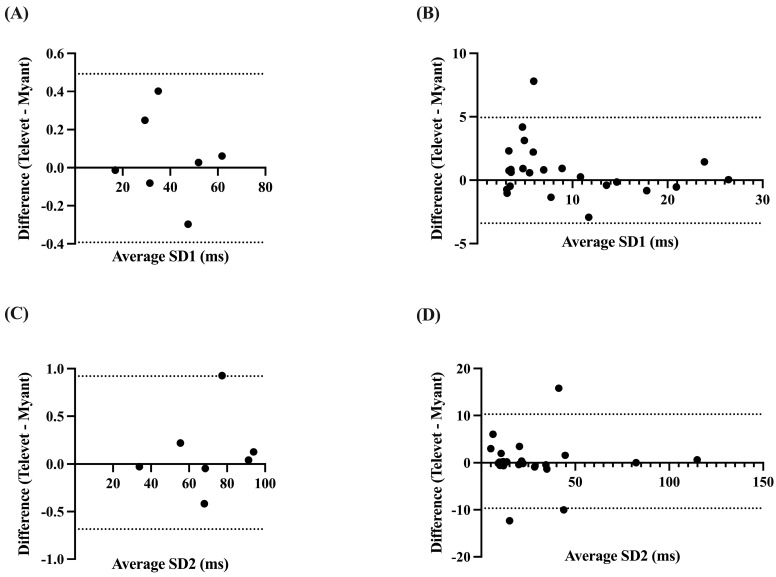
Bland–Altman plots demonstrating the agreement between the two devices for nonlinear HRV parameters at rest (**A**,**C**) and during exercise (**B**,**D**). SD1 = Poincaré plot standard deviation perpendicular to the line of identity; SD2 = Poincaré plot standard deviation along the line of identity. X-axes are the average of SD1 or SD2 for each device; y-axes are the difference of Televet minus Myant SD1 or SD2. Solid lines indicate bias and dashed lines indicate 95% limits of agreement.

**Table 1 animals-13-00512-t001:** Heart rate variability metrics obtained from electrocardiograms using a smart textile (Myant) and standard telemetric device (Televet) during rest and exercise (pre-exercise walk, trot, canter, post-exercise walk).

Metric	Rest		Pre-Exercise Walk		Trot		Canter	Post-Exercise Walk		
Device	Myant	Televet	Myant	Televet	Myant	Televet	Myant	Televet	Myant	Televet
Mean HR (bpm)	42.0 ± 4.7	42.0 ± 4.7	56.7 ± 4.4	56.6 ± 4.3	105.9 ± 8.7	106.2 ± 8.5	129.5 ± 9.2	130.5 ± 8.6	103.4 ± 5.2	101.2 ± 4.7
SDRR (ms)	56.8 ± 17.6	56.8 ± 17.6	40.1 ± 23.2	41.5 ± 23.0	14.9 ± 11.9	14.1 ± 8.7	7.4 ± 5.1	7.1 ± 2.2	12.5 ± 5.3	12.7 ± 5.4
HRV TI	11.9 ± 3.5	11.5 ± 4.5	7.0 ± 2.1	7.0 ± 2.0	4.4 ± 2.2	4.4 ± 2.0	2.8 ± 1.2	3.6 ± 2.7	3.4 ± 1.3	3.5 ± 1.1
TIRR (ms)	271.6 ± 82.6	271.4 ± 78.5	144.9 ± 65.4	122.0 ± 74.4	59.8 ± 37.6	61.9 ± 21.4	35.3 ± 19.9	51.8 ± 49.7	53.3 ± 22.3	51.2 ± 19.6
RMSSD (ms)	55.2 ± 21.7	55.2 ± 21.6	25.5 ± 7.7	25.5 ± 7.9	8.8 ± 5.8	13.3 ± 13.4	6.7 ± 5.9	9.0 ± 4.4	6.6 ± 2.7	7.3 ± 2.3
SD1 (ms)	39.1 ± 15.4	39.2 ± 15.3	18.3 ± 5.6	18.3 ± 5.8	6.3 ± 4.1	9.8 ± 9.4	4.8 ± 4.2	6.4 ± 3.1	4.7 ± 1.9	5.2 ± 1.7
SD2 (ms)	69.7 ± 20.8	69.9 ± 20.9	52.4 ± 33.5	54.7 ± 32.8	21.2 ± 15.0	33.9 ± 38.2	10.0 ± 6.6	17.1 ± 19.2	16.8 ± 7.2	17.0 ± 7.4

HR = heart rate; SDRR = standard deviation of R-R intervals; HRV TI = heart rate variability triangular index; TIRR = triangular interpolation of R-R interval histogram; RMSSD = root mean square of successive R-R interval differences; SD1 = Poincaré plot standard deviation perpendicular to the line of identity; SD2 = Poincaré plot standard deviation along the line of identity.

**Table 2 animals-13-00512-t002:** Correlation coefficient and associated *p* value of heart rate variability metrics obtained from two different devices (smart textile—Myant, standard telemetric—Televet) during rest and exercise.

Metric	Rest	Pre-Exercise Walk	Trot	Canter	Post-Exercise Walk
Mean HR (bpm)	*r* = 1.0 (*p* < 0.0001)	*r* = 1.0 (*p* < 0.0001)	*r* = 1.0 (*p* < 0.0001)	*r* = 1.0 (*p* < 0.0001)	*r* = 0.91 (*p* = 0.0007)
SDRR (ms)	*r* = 1.0 (*p* < 0.0001)	*r* = 0.99 (*p* < 0.0001)	*r* = 0.99 (*p* = 0.0007)	*r* = 0.99 (*p* = 0.006)	*r* = 0.97 (*p* = 0.0008)
HRV TI	*r* = 0.81 (*p* = 0.01)	*r* = 0.73 (*p* = 0.03)	*r* = 0.98 (*p* = 0.0004)	*r* = 0.75 (*p* = 0.03)	*r* = 0.98 (*p* = 0.0002)
TIRR (ms)	*r* = 0.98 (*p* < 0.0001)	*r* = 0.82 (*p* = 0.01)	*r* = 0.85 (*p* = 0.03)	*r* = 0.78 (*p* = 0.04)	*r* = 0.97 (*p* = 0.0008)
RMSSD (ms)	*r* = 1.0 (*p* < 0.0001)	*r* = 0.99 (*p* < 0.0001)	*r* = 0.94 (*p* = 0.003)	*r* = 0.92 (*p* = 0.005)	*r* = 0.87 (*p* = 0.01)
SD1 (ms)	*r* = 1.0 (*p* < 0.0001)	*r* = 0.99 (*p* < 0.0001)	*r* = 0.92 (*p* = 0.005)	*r* = 0.92 (*p* = 0.005)	*r* = 0.88 (*p* = 0.01)
SD2 (ms)	*r* = 1.0 (*p* < 0.0001)	*r* = 0.98 (*p* < 0.0001)	*r* = 0.99 (*p* = 0.0004)	*r* = 0.99 (*p* = 0.0005)	*r* = 0.97 (*p* = 0.0005)

HR = heart rate; SDRR = standard deviation of R-R intervals; HRV TI = heart rate variability triangular index; TIRR = triangular interpolation of R-R interval histogram; RMSSD = root mean square of successive R-R interval differences; SD1 = Poincaré plot standard deviation perpendicular to the line of identity; SD2 = Poincaré plot standard deviation along the line of identity.

## Data Availability

Data are available from authors upon reasonable request.

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
