# Peer review of "Validation of an Equine Smart Textile System for Heart Rate Variability: A Preliminary Study"

_animals, 2023, doi:10.3390/ani13030512_

Round 1
Reviewer 1 Report
The manuscript submitted by the authors aimed at validating a smart textile system against the gold standard telemetric device to record Heart Rate (HR) and Heart Rate Variability (HRV) in horses at rest and during exercise.
To this aim, electrocardiograms were obtained using both devices simultaneously in seven horses at rest and during exercise. Results shown that the two devices are in agreement concerning the ability to record HR and HRV during rest and submaximal exercise in healthy horses, supporting that the textile system is a reliable and user-friendly alternative device for ECG monitoring of horses during rest and exercise. Moreover, the fact that the device does not require hair preparation is considered a technical advantage.
Overall, the manuscript is well-written, hypothesis and objectives are clearly stated and the methods used allow to fulfil their goals. Results are clearly presented and discussion is exhaustive.
Minor :
1. a few clarificarion on the clinical meaning of the three-domain metrics studied would be of value for a non-cardiology-expert reader.
2. the Authors refer to the presence of artifacts as a flaw of the gold standard method studied (televet) and as a reason to test the reliability fo the new system but no indication are reported on this aspects in the reults or in the discussion. Clarifications on this aspects should be provided given the statement made in the introduction.
L. 67: please verify "confuctive". Shall this be "conductive"?
L. 70-71: it is unclear the reference to the recording device here and its role in signal quality. please check.
Author Response
The manuscript submitted by the authors aimed at validating a smart textile system against the gold standard telemetric device to record Heart Rate (HR) and Heart Rate Variability (HRV) in horses at rest and during exercise.
To this aim, electrocardiograms were obtained using both devices simultaneously in seven horses at rest and during exercise. Results shown that the two devices are in agreement concerning the ability to record HR and HRV during rest and submaximal exercise in healthy horses, supporting that the textile system is a reliable and user-friendly alternative device for ECG monitoring of horses during rest and exercise. Moreover, the fact that the device does not require hair preparation is considered a technical advantage.
Overall, the manuscript is well-written, hypothesis and objectives are clearly stated and the methods used allow to fulfil their goals. Results are clearly presented and discussion is exhaustive.
Minor :
a few clarificarion on the clinical meaning of the three-domain metrics studied would be of value for a non-cardiology-expert reader.
Thank you, we have added more information to line 138: Additionally, nonlinear measurements were obtained from the Poincaré plot by fitting an ellipse to the plotted points. The standard deviation perpendicular to the line of identity (SD1) and the standard deviation along the line of identity (SD2) of the Poincaré plot were determined, where SD1 correlates with short-term HRV and SD2 correlated with long-term HRV [9,12,21].
the Authors refer to the presence of artifacts as a flaw of the gold standard method studied (televet) and as a reason to test the reliability fo the new system but no indication are reported on this aspects in the reults or in the discussion. Clarifications on this aspects should be provided given the statement made in the introduction.
We have clarified in both the introduction and discussion that the goal of this study was to compare HRV metrics, and not to quantify MAs. Please see lines 73-78: This study utilized the same recording device without a gold standard device, with comparisons only made between electrode types and data were collected only from horses at rest, without any structured exercise. Furthermore, no metrics of HRV were assessed. Therefore, the objective of this study was to validate a smart textile system against the gold standard telemetric device for measures of HRV in horses at rest and during submaximal exercise.
Lines 373-377: More recently, we have observed rates of MAs to be below 0.5% in unrestrained horses at rest using a different smart textile system, indicating that textile electrodes may be less prone to MAs [18]. We did not quantify MAs in the present study, as the aim was to compare measures of HRV. Further work on the presence of MAs obtained using textile systems, particularly during exercise, is warranted.
- 67: please verify "confuctive". Shall this be "conductive"?
Thank you, we have corrected this typo.
- 70-71: it is unclear the reference to the recording device here and its role in signal quality. please check.
Thank you, we have addressed this in our correction above. Please see lines 73-78.
Reviewer 2 Report
Revision of the article entitled “Validation of an equine smart textile system for electrocardiogram recording: a preliminary study”
Authors compared data of heart rate variability obtained using a smart textile system (Myant) to the gold standard telemetric device (Televet) both at rest and during exercise. Results showed any significant differences in metrics of heart rate or heart rate variability, indicating that the smart textile system is a reliable alternative device for ECG monitoring of horses.
Line 74: correct “hroses”
Line 80-82: add the sentence regarding the Informed Consent Statement (line274-275) here too
Line 83: add a figure of the Myant and a figure of a horse dressed with it
Line 111: How long was the post exercise recording?
Line 128: add the classification of the correlation in accordance to Pearson coefficient
Line 136-139: did he not tolerate the device or was he simply a hot horse?
Figure 1: I would put myant and televet below and above by selecting exactly the same beginning and end of the track. In this way the concordance between the two devices is immediately evident but also the division between the two is understood well.
Table 1 : I think that data of HR pre-exercise televet is wrong
Line 156: why didn't you analyze the different exercise phases individually?
Figure 2: Myant and Televet are the two axes. For each point the abscissa indicates the value of Myant and the ordinate that of Televet. The legend of myant circle and televet square is wrong
Results: there are no results of the t-test taht you mentioned in the materials and methods
Line 207: you carried out the analyzes by considering all the exercise phases together, not individually
Author Response
Revision of the article entitled “Validation of an equine smart textile system for electrocardiogram recording: a preliminary study”
Authors compared data of heart rate variability obtained using a smart textile system (Myant) to the gold standard telemetric device (Televet) both at rest and during exercise. Results showed any significant differences in metrics of heart rate or heart rate variability, indicating that the smart textile system is a reliable alternative device for ECG monitoring of horses.
Line 74: correct “hroses”
Thank you, we have corrected that typo.
Line 80-82: add the sentence regarding the Informed Consent Statement (line274-275) here too
Thank you, we have added that information.
Line 83: add a figure of the Myant and a figure of a horse dressed with it
We agree with the reviewer that this would be beneficial for readers and have added this figure. Please see figure 1.
Line 111: How long was the post exercise recording?
We have added this detail, please see line 119: Horses were walked in hand prior to and completion of the standardized exercise test to obtain data for walk and post-exercise walk (active recovery) for 5 min each.
Line 128: add the classification of the correlation in accordance to Pearson coefficient
Thank you, we have added this information to lines 147-150: A Pearson correlation coefficient was calculated for both devices, where an r value of 1.0 was considered to be a perfect correlation, an r vale of 0.8 to 1.0 was considered to be a very strong correlation, and an r value of 0.6 to 0.8 was considered to be a strong correlation.
Line 136-139: did he not tolerate the device or was he simply a hot horse?
He was simply a hot horse and tolerated both devices well at rest and at a walk, but became excited at the trot and canter. We have clarified this in lines 156-158: One horse’s data was excluded from the exercise portion due to overexcitement and bucking on the lunge line, which did not appear to be due to either device (n = 6).
Figure 1: I would put myant and televet below and above by selecting exactly the same beginning and end of the track. In this way the concordance between the two devices is immediately evident but also the division between the two is understood well.
We agree with the reviewer and have updated the figure accordingly. Please see figure 2.
Table 1 : I think that data of HR pre-exercise televet is wrong
Thank you, we have corrected that error.
Line 156: why didn't you analyze the different exercise phases individually?
We analyzed the different phases individually, as shown in table 1. However, we only reported statistics for pooled activities. We agree with the reviewer that this is valuable information and as per the reviewer’s recommendation, we have added an additional table (table 2) that lists the statistics for each activity. We have left the figures as pooled data for clarity.
Figure 2: Myant and Televet are the two axes. For each point the abscissa indicates the value of Myant and the ordinate that of Televet. The legend of myant circle and televet square is wrong
Thank you, we have corrected that.
Results: there are no results of the t-test taht you mentioned in the materials and methods
Thank you, we have added a statement that no differences were found (lines 164-165): Paired t-tests revealed no significant differences between the two devices for any of the metrics assessed for individual activities.
We have also added p values from the t-tests throughout the results.
Line 207: you carried out the analyzes by considering all the exercise phases together, not individually
We have left this statement in, as the results for individual activities have been added.
Reviewer 3 Report
This study by McCrae et al. sought to validate an equine smart textile system for ECG recording. The study is well written and executed. However, the scientific contribution of the study is very limited.
Reviewer 4 Report
This paper validated a user-friendly, smart textile system (Myant) against the gold standard telemetric device (Televet) in measuring HR and HRV data. A very strong correlation was found between the two devices for HR and HRV parameters. This study demonstrates that a smart textile system is reliable for the assessment of HR and HRV of horses at rest and during submaximal exercise.
As a part of a series study, the author proved that this smart textile system from Myant Inc. could be used in practice. Scientific development dramatically changed the veterinary practice. Although these smart textile systems did not show more exciting advantages, the accessible and comfortable of these systems encourage veterinarians to monitor equine ECG routinely. The authors are encouraged to explore the role of HRV in equine cardiovascular pathophysiology in the future.
Round 2
Reviewer 2 Report
Thanks to the authors for the replies. The revised version has improved the quality of the paper and I think it is so good for publication.